# Genomic Analysis of the Deep-Sea Bacterium *Shewanella* sp. MTB7 Reveals Backgrounds Related to Its Deep-Sea Environment Adaptation

**DOI:** 10.3390/microorganisms11030798

**Published:** 2023-03-21

**Authors:** Sicong Li, Jiahua Wang, Jie Liu, Hongcai Zhang, Tianqiang Bao, Chengwen Sun, Jiasong Fang, Junwei Cao

**Affiliations:** 1Shanghai Engineering Research Center of Hadal Science and Technology, College of Marine Sciences, Shanghai Ocean University, Shanghai 201306, Chinajsfang@shou.edu.cn (J.F.); 2Laboratory for Marine Biology and Biotechnology, Qingdao National Laboratory for Marine Science and Technology, Qingdao 266237, China

**Keywords:** deep sea, *Shewanella*, psychrophilic, complete genome

## Abstract

*Shewanella* species are widely distributed in various environments, especially deep-sea sediments, due to their remarkable ability to utilize multiple electron receptors and versatile metabolic capabilities. In this study, a novel facultatively anaerobic, psychrophilic, and piezotolerant bacterium, *Shewanella* sp. MTB7, was isolated from the Mariana Trench at a depth of 5900 m. Here, we report its complete genome sequence and adaptation strategies for survival in deep-sea environments. MTB7 contains what is currently the third-largest genome among all isolated *Shewanella* strains and shows higher coding density than neighboring strains. Metabolically, MTB7 is predicted to utilize various carbon and nitrogen sources. D-amino acid utilization and HGT-derived purine-degrading genes could contribute to its oligotrophic adaptation. For respiration, the cytochrome *o* ubiquinol oxidase genes *cyoABCDE,* typically expressed at high oxygen concentrations, are missing. Conversely, a series of anaerobic respiratory genes are employed, including fumarate reductase, polysulfide reductase, trimethylamine-N-oxide reductase, crotonobetaine reductase, and Mtr subunits. The glycine reductase genes and the triplication of dimethyl sulfoxide reductase genes absent in neighboring strains could also help MTB7 survive in low-oxygen environments. Many genes encoding cold-shock proteins, glycine betaine transporters and biosynthetic enzymes, and reactive oxygen species-scavenging proteins could contribute to its low-temperature adaptation. The genomic analysis of MTB7 will deepen our understanding of microbial adaptation strategies in deep-sea environments.

## 1. Introduction

Hadal trenches are the deepest areas of the ocean as well as one of the least-explored biospheres on Earth [1,2]. Microbial abundance and diversity below 6000 m depth were thought to be extremely low due to the low temperature, high hydrostatic pressure (HHP), and oligotrophic and low-oxygen environment in hadal trenches [3,4]. And yet various piezophilic, piezotolerant and anaerobic bacteria, have been reported to thrive in this deep biosphere. However, there is still limited information on the adaptation strategies of these microbes that are capable of living in the hadal trenches [5,6].

The genus *Shewanella* comprises a plethora of species that are widely distributed in a variety of environments, especially in marine and deep-sea sediments, due to their remarkable ability to utilize multiple electron receptors and versatile metabolic capabilities [7,8,9]. Many *Shewanella* species have been isolated from diverse deep-sea environments [10,11,12,13]. For example, two *Shewanella benthica* strains, KT99 and DB21MT-2, were isolated from the Kermadec and Mariana Trenches at depths of 9856 m and 10,898 m, respectively [14,15], and another strain, *S. violacea* DSS12, was isolated from abyssopelagic sediment in the Ryukyu Trench at a depth of 5110 m [16].

Many *Shewanella* species exhibit adaptability to high hydrostatic pressure (piezophilic or piezotolerant, e.g., *S. psychrophila* WP2 and *S. piezotolerans* WP3) and low temperature (psychrophilic, e.g., *S. psychropiezotolerans* YLB-06, *S. eurypsychrophilus* YLB-08, and *S.* sp.YLB-09) [12,17]. In addition, many *Shewanella* species can use various terminal electron acceptors, such as trimethylamine N-oxide (TMAO), DMSO, sulfur, Fe (III), nitrite, and nitrate [18]. These adaptation strategies and metabolic plasticity are believed to be the reasons that enable them to survive in the hadal trenches.

In this study, we isolated a *Shewanella* strain, MTB7, from the water-sediment interface of the Mariana Trench at a depth of 5900 m (142.3° E, 11.6° N) using deep-sea sampling lander, which is currently the third-deepest isolation of this genus. The strain shows psychrophilic characteristics with an optimal temperature of 4–10 °C and highest growth at a temperature of 20 °C. Strain MTB7 currently preoccupies the third-largest genome size among all isolated *Shewanella* species. In this paper, we report the complete genome sequence of strain MTB7 and provide additional information for further understanding of its genome expansion and the possible adaptive strategies of the *Shewanella* genus to thrive in oligotrophic, low-oxygen, and low-temperature environments.

## 2. Materials and Methods

### 2.1. Sample Description and Strain Isolation

Water-sediment interface samples were collected at a depth of 5900 m (142.3° E, 11.6° N) in the Mariana Trench during a cruise aboard R/V “ZHANG JIAN HAO” in December 2016. Niskin bottles fitted on a hadal lander (Shanghai Ocean University, China) were used to collect samples from the trench. Two hours after the lander reached the seafloor, samples were collected using the Niskin bottles, and the lander was recovered. After recovering on board, the water-sediment interface samples were immediately subsampled and stored in 50-mL sterile centrifuge tubes in a cold room at 4 °C on board the ship.

An amount of 0.5 mL of the water-sediment interface sample was diluted into 500 mL of autoclave seawater and was then coated on culture plates with marine agar 2216 (MA, BD Difco) at 10 °C. After 7 days, the microbial colonies were picked and purified by the standard dilution plating technique with the same culture condition. The temperature for optimal growth was tested at 0, 4, 10, 16, 20, 25, 28 and 30 °C in duplicates. Stock cultures were stored at −80 °C with 20% (*v/v*) glycerol.

### 2.2. DNA Extraction, Genome Sequencing, and Assembly

Cells of strain MTB7 were cultured in marine broth 2216 (MB, BD) at 10 °C and 200 rpm for 48 h. The genomic DNA of strain MTB7 was extracted using the Qiagen DNA extraction kit. Total DNA was subjected to quality control by gel electrophoresis and quantified by a Qubit fluorometer (Thermo Fisher Scientific, Waltham, MA, USA). The complete genome sequence of strain MTB7 was sequenced at Nextomics Biosciences Co., Ltd. (Wuhan, PR China), using both Illumina and Nanopore sequencing technologies. For Illumina sequencing, 500 bp paired-end libraries were prepared, and paired-end reads (3,723,311 × 150 bp reads) were generated on the Illumina Hiseq X Ten machine. Illumina raw reads were quality filtered and adapter trimmed with Trimmomatic v0.36 [19]. Oxford Nanopore Technologies (ONT) sequencing libraries were prepared using the manufacturer’s ligation sequencing kit (SQK-LSK109), and sequencing was carried out on a PromethION device using flow cells (FLO-PRO002) without fragmentation. Illumina and Nanopore reads were hybrid de novo assembled using Canu (version 1.7) in the normal assembly mode. Gene annotations were performed by the NCBI prokaryotic genome annotation pipeline (PGAP) [20]. The complete genome sequence of *Shewanella* sp. strain MTB7 was deposited in the GenBank database under the accession number CP085636.

### 2.3. Gene Annotation and Metabolic Pathway Reconstruction

The NCBI prokaryotic genome annotation pipeline was used in ORF prediction and gene annotation [20]. Gene functional categories were identified by a BLASTp search in the Clusters of Orthologous Groups (COG) database [21]. Metabolic pathway analysis was performed by searching the KEGG GENES database with BlastKOALA [22]. The genomic islands were predicted using IslandViewer 4 [23]. The transporters were predicted by BLASTp against TransportDB 2.0 [24].

### 2.4. Phylogenetic Analysis

The 120 conserved bacterial marker genes of the GTDB taxonomy were used to study the phylogeny of strain MTB7 and its related strains. The sequences of 120 concentrated proteins in the genomes were predicted using GTDB-Tk (database version: Release 07-RS207) [25] and separately aligned using Clustal Omega [26]. All aligned sequences were manually degapped and tandemly connected (5038 amino acid residue in total after degap). The phylogenetic tree was constructed using FastTree2 with the neighbor-joining method [27], and a bootstrap analysis with 1000 replicates was performed to assess the robustness of the tree. Finally, the phylogenetic tree was plotted using iTOL [28].

### 2.5. Genomic Comparisons

The protein families of strain MTB7 and phylogenetically related strains were clustered using a local OrthoMCL 2.0.9 [29] with the following cut-off values: identity, 50%; query coverage, 50%; E-value, 1E-10; score, 40; and MCL Markov clustering inflation index, 1.5. The protein families employed by only one strain were considered to be strain specific. Average nucleotide identity (ANI) and average amino acid identity (AAI) were calculated using fastANI [30] and CompareM (https://github.com/dparks1134/CompareM, accessed on 16 November 2022), respectively, with default parameters.

### 2.6. Provirus Prediction and Classification

First, VirSorter2 (v2.2.2) [31] was used to predict viral sequences in strain MTB7 with the setting “–include-groups dsDNAphage, ssDNA –min-length 5000 –min-score 0.5.” The mode of searching double-stranded DNA (dsDNA) was appropriate for recovery of temperate viruses, and the minimum score (0.5) was chosen for maximal sensitivity. Second, CheckV (v0.8.1) [32] was used to control the quality of the viral sequences. Third, the CheckV-trimmed sequences were passed through VirSorter2 again and formatted to serve as input to DRAM-v (v1.2.4) [33] for viral annotation. Retained viral sequences were curated according to the widely recognized empirical screening criteria based on the information of viral and host gene counts, score, hallmark gene counts, and sequence length [34]. The classification of proviruses followed the online protocol (https://www.protocols.io/view/viral-sequence-identification-sop-with-virsorter2-5qpvoyqebg4o/v2, accessed on 16 November 2022).

## 3. Results and Discussion

### 3.1. The Phylogeny of Strain MTB7

The 16S rRNA sequences showed that strain MTB7 was closely related to *Shewanella hanedai* CIP 103207^T^, *Shewanella woodyi* ATCC 51908^T^, and *Shewanella sediminis* HAW-EB3^T^, with the identities of 97.93%, 97.29%, and 97.14%, respectively. However, many *Shewanella* species contain heterogenetic 16S rRNA sequences, e.g., *S. psychropiezotolerans* YLB-06 contains 12 16S rDNA sequences with 11 sequence variants. Therefore, 16S rDNA sequence-based phylogenetic analysis could lead to a bias in the phylogeny of strain MTB7. In this study, we obtained the complete genome sequences of strain MTB7 and constructed a phylogenetic tree based on 120 conserved protein sequences (known as GTDB taxonomy), and a subtree including strain MTB7 and its 12 neighboring strains is displayed in Figure 1. Clearly, strain MTB7 belongs to genus *Shewanella* and is closely related to *S. woodyi* ATCC 51908, which was isolated from seawater at a depth of 200 m in the Alboran Sea [35]. 

### 3.2. The Genomic Features of Strain MTB7

The complete genome of strain MTB7 consists of one chromosome with a total length of 6,367,435 base pairs (bp), which is smaller than *Shewanella* sp. YLB-07 (7,295,150 bp) and *S. psychropiezotolerans* YLB-06 (6,449,204 bp) but larger than the other 250 published genomes of *Shewanella* strains. The G+C content of the MTB7 genome is 42.70%. The average nucleotide identity (ANI) and average amino acid identity (AAI) between strain MTB7 and *S. woodyi* ATCC 51908 are 81.07% and 84.84%, respectively. The genome of strain MTB7 contains 5421 genes, including 5301 protein-coding genes, 60 tRNAs, 4 ncRNAs, 49 pseudogenes and 11 rRNA operons (Table 1). None of the eleven 16S rDNA sequences showed 100% identity with each other. Upon COG classification, 3953 (75.57%) protein-coding genes were assigned to 22 categories (Appendix A). The major COG categories were signal transduction mechanisms (COG-T, 7.73%), amino acid transport and metabolism (COG-E, 7.67%), transcription (COG-K, 7.53%), energy production and conversion (COG-C, 7.47%), and general function prediction only (COG-R, 7.40%). The graphical representation of the MTB7 genome is shown in Figure 2.

To study the reasons for genome expansion in strain MTB7, the Markov clustering algorithm OrthoMCL was used to identify orthologous gene families, as well as paralogous genes of each *Shewanella* genome, which reflected their genomic redundancy. On one hand, strain MTB7 has more orthologous families (5198) than neighboring strains (from 3418 to 5070) (Appendix A). Moreover, genomic islands (GIs) are parts of genomes that have evidence of horizontal origins, and as many as 18 genomic islands were predicted in strain MTB7 (Appendix A). In addition, 29 transposases, 5 recombinases, 7 site-specific integrases, and 10 tyrosine-type recombinases/integrases were identified in strain MTB7, of which 38 (74.51%) were absent in *S. woodyi* ATCC 51908 (Appendix A). In addition, seven proviruses were predicted in the MTB7 genome, of which three were high quality or complete (Appendix A). Among the 307 viral protein sequences, 157 (51.14%) have no homolog in the neighboring *Shewanella* species of strain MTB7 (query coverage, 50% and E-value, 1E-3). These pieces of evidence suggest that horizontal gene transfer (HGT) has contributed to the increase of orthologs and genome expansion of strain MTB7. On the other hand, duplication within a genome leads to paralogous genes, and the number of paralogous genes could reflect the redundancy of a genome. We found that the 117 complete genomes of *Shewanella* strains possess 130 paralogous genes within 90 gene families on average, whereas strain MTB7 exhibits 338 paralogous genes within 225 families (BLASTp identity, 50%; query coverage, 50; E-value, 1E-5, and MCL inflation, 1.5), suggesting that high genomic redundancy could also be one of the reasons behind its genome expansion (Appendix A).

Interestingly, although strain MTB7 has a smaller genome size than *S. psychropiezotolerans* YLB-06, we noticed that it possesses more active (non-pseudo) genes than strain YLB-06 (5255) (Appendix A). Further study showed that the coding density in strain MTB7 (84.86%) is higher than in any closely related strains, e.g., 79.61% in *S. violacea* DSS12, 77.28% in YLB-06, and 78.57% in *S. benthica* DB21MT-2 (Appendix A). This result suggests that an increase in genomic coding density could be an evolutionary strategy for its deep-sea adaptation. Generally, microbial adaptation to the oligotrophic environments could be promoted by increasing metabolic potentials without increasing the energetic burden of genome replication.

### 3.3. Reconstruction of Metabolic Pathways

To study how strain MTB7 adapts to the deep marine water-sediment interface, described as oligotrophy and low oxygen concentrations [36,37], the metabolic pathways of strain MTB7 and the related strain *S. woodyi* ATCC 51908 were reconstructed (Figure 2) using BlastKOALA and compared with each other. Notably, to avoid the biases of annotation between the two genomes, (1) any protein with a strain-specific KO number must be from a strain-specific orthologous family that is not possessed by the other strain (identity, 30%; query coverage, 50%; E-value, 1E-3; and MCL inflation, 1.5). In addition, (2) all reference sequences with such KO numbers were downloaded from the KEGG database and aligned with all proteins of the other genome to confirm the absence of these KO numbers in the other genome (identity, 30%; query coverage, 50%; and E-value, 1E-3).

In the aspect of carbon utilization, strain MTB7 and *S. woodyi* were largely different. There were 17 genes of carbohydrate transporters predicted in strain MTB7, of which 13 were specific to strain MTB7, including ribose ABC transporters and L-fucose: H^+^ symporters. The two strains shared the pathway of fructose utilization, whereas the genes for glucose, sucrose, starch, cellodextrin, cellobiose and trehalose utilization were only identified in *S. woodyi*. The different capabilities of carbohydrate utilization between strain MTB7 and *S. woodyi* were confirmed by culture experiments (Appendix A). This suggests that MTB7 lost many carbohydrate-utilizing genes in the deep-sea habitat where carbohydrates were scarce. Conversely, there were various genes involved in amino acid utilization that were predicted in MTB7, including 37 amino acid/peptide transporters. A high proportion of D-amino acids was suggested to indicate old and refractory organic material [38]. Notably, the genes of D-serine transporter (HWQ47_08850), D-serine ammonia-lyase (HWQ47_08855), and D-amino acid dehydrogenase (HWQ47_23045) were only predicted in MTB7, suggesting that MTB7 could survive with degradation of refractory organic compounds enriched in deep marine environments (Appendix A). Moreover, phospholipid transporters were predicted, indicating that phospholipids could also be a possible carbon source for MTB7. In addition, the complete purine degradation pathways, as well as four strain-specific genes of xanthine, adenine/guanine/hypoxanthine, and uric acid permeases were only found in MTB7 and not in *S. woodyi* (Appendix A). Among these purine transporters, HWQ47_21770 (purine permease) and HWQ47_12615 (NCS2 family permease) showed the highest identities with *Shewanella marina* (with identity of 84.89%) and *Shewanella hanedai* (with identity of 93.88%), respectively. However, the other top 10 hits were from *Protobacterium* (with identities of 76.0% to 78.167%) and *Moritella* (with identities of 85.94% to 87.98%), respectively. Considering that *S. marina* and *S. hanedai* are not the neighboring species of strain MTB7, we proposed that acquisition of purine utilizing genes via HGT could also contribute to strain MTB7 survival in oligotrophic environments.

For nitrogen metabolism, ammonia, urea, ethanolamine, and putrescine were putatively utilized by strain MTB7. Furthermore, both strain MTB7 and *S. woodyi* were predicted to have genes for denitrification (including *napAB*, *nirK,* and *norBC*) and dissimilatory nitrate reduction (including *napAB*, *nirBD,* and *nrfAH*) (Appendix A), although the membrane type nitrate reductase (encoded by *narGHI)* was only predicted in *S. woodyi.* The ability to utilize multiple carbon and nitrogen sources may be one of the important strategies for its adaptation to the oligotrophic environment.

### 3.4. Multiple Electron Transfer Chains

Respiration is a central metabolism of many microorganisms. A series of respiratory chain complexes, I, II, III, IV, and V, were found in strains MTB7 and *S. woodyi*. For aerobic respiration, MTB7 contained cytochrome c oxidase (encoded by *coxABC*), as well as cytochrome bd ubiquinol oxidase (encoded by *cydAB*), which is involved in aerobic respiration with low oxygen concentration [39]. However, the cytochrome o ubiquinol oxidase genes *cyoABCDE* were identified in *S. woodyi* and neighboring strains but not in strain MTB7. Considering that *cyo* genes were typically expressed at high oxygen concentrations [39], we posit that a strain-specific gene loss in MTB7 could be an energy-saving strategy for its survival in deep water-sediment interface with both low oxygen concentration and limited nutrients.

In addition to genes for aerobic respiration, many genes for anaerobic respiration were found in MTB7, including *frdABC*, *mtrABC*, *grdABCD, caiABCDE*, *psrABC, torCAD,* and *dmsABD* (Appendix A). *frdABC* encoded fumarate reductase subunits, which are homologs of succinate dehydrogenase (Sdh) and catalyze fumarate to succinate. Different from the *sdh* genes involved in aerobic respiration, the *frd* genes are involved in anaerobic respiration and are induced when the oxygen concentration is close to zero [40].

Glycine reductase is present in various anaerobic bacteria that utilize glycine as an electron acceptor in Stickland reactions [41,42]. In addition to glycine, *E. acidaminophilum* can utilize sarcosine (N-methylglycine) and betaine (N,N,N-trimethylglycine) as electron acceptors, if an electron donor such as formate is additionally provided [42,43,44]. A gene operon was identified in strain MTB7 that encoded glycine reductase complex component B subunit gamma (*grdB*), subunits alpha and beta (*grdE*), selenide, water dikinase (*selD*), betaine reductase complex component C subunit alpha (*grdD*), subunit beta (*grdC*), betaine reductase complex component A (*grdA*), thioredoxin 1 (*trxA*), and NADPH-dependent thioredoxin reductase (*trxB*). In addition, the genes of L-seryl-tRNA (Ser) seleniumtransferase (*selA*) and selenocysteine-specific elongation factor (*selB*) were identified in the upstream of the glycine reductase operon, which might participate in the biogenesis of selenoprotein GrdA (Figure 3). Notably, these genes showed >70% identities with those from *Photobacterium* species but were not identified in other neighboring *Shewanella* genomes (Figure 1), suggesting that horizontal gene transfer has contributed to the adaptation of strain MTB7 to anoxic deep-marine water-sediment interface. In addition, strain MTB7 also harbors the strain-specific gene operon of crotonobetaine reductase (encoded by *caiABCDE*) compared to neighboring strains (Figure 3), which is an inducible enzyme detectable only in cells grown anaerobically in the presence of L(-)-carnitine or crotonobetaine as electron acceptors [45].

Dimethyl sulfoxide (DMSO) reductase in *S. oneidensis* exhibited menaquinol-DMSO oxidoreductase activity, which reduced DMSO to dimethyl sulfide (DMS) [46], and was induced in response to a fall in oxygen tension. Moreover, Xiong et al. (2016) reported that deletion of DMSO respiratory genes severely impaired the growth of *Shewanella piezotolerans* WP3 at high pressure and/or low temperature [47]. In strain MTB7, two gene clusters were identified encoding DMSO reductase subunits (encoded by *dmsA/B/D*) and MtrB/PioB family proteins. One *dms* cluster contains a single operon (HWQ47_02440 to HWQ47_02460) and showed a similar pattern (*dmsE-mtrB-dsmABD*) with those from neighboring strains (e.g., SWOO_RS01180 to SWOO_RS01200 in *S. woodyi*). The other *dms* cluster (HWQ47_25505 to HWQ47_25540) is located next to the menaquinone biosynthesis genes (*menDHCE*) and contains two *dms* operons, upstream *dmsE-mtrB-dsmABD* and downstream *mtrB-dsmABDAB* (Figure 3), both of which were missing in *S. woodyi.* Interestingly, the three *dsmA* genes in this cluster showed high similarity to each other, as well as to the three *dmsB* and two *dmsD* genes, suggesting that local genomic duplication also played a role in the adaptation of strain MTB7 to hadal water-sediment interface with combined high pressure, low temperature, and oxygen deprivation.

In addition, strain MTB7 also encoded polysulfide reductase (encoded by *psrABC*), trimethylamine-N-oxide (TMAO) reductase (encoded by *torACD*), and Mtr subunits (encoded by *mtrABC*), which were reported to be involved in anaerobic respiration on S_0_ or S_2_O_3_^2−^, TMAO, and extracellular mineral metal oxides (e.g., Fe(III)) as electron acceptors, respectively [48,49,50,51,52,53], which also contribute to its survival in low-oxygen water-sediment interface.

### 3.5. Genes Involved in Low-Temperature Adaptation

Strain MTB7 shows psychrophilic characteristics with optimal temperature of 4–10 °C and highest growth temperature of 20 °C. We surveyed its genome and identified many genes putatively involved in cold adaptation (Appendix A). It is generally believed that the mechanism of action of cold shock proteins (CSPs) is similar to that of an RNA molecular chaperone, combining with single-stranded ribonucleic acid/deoxyribonucleic acid and promoting the effective folding of ribonucleic acid/deoxyribonucleic acid, preventing protein misfolding, and ensuring the stability of ribonucleic acid and deoxyribonucleic acid secondary structures [54]. In the genome of strain MTB7, six CPSs were identified, of which one showed high similarity to that of *Photobacterium profundum* SS9, a model psychrophilic and piezophilic Gram-negative bacteria.

Glycine betaine is a kind of compatible solute involved in osmoprotection and salt stress and high pressure (HHP) protection and has been found to be accumulated in deep-sea bacteria at low temperatures, which could help stabilize enzymes, DNA, and cytoplasmic membranes [55,56,57]. Although the reported genes of other osmoprotectants (e.g., ectoine, polyhydroxyalkanoate, and trehalose) were not found in strain MTB7, it harbors as many as eight genes of BCCT transporters of glycine betaine, in addition to glycine betaine/L-proline ABC transporters (encoded by *proVWX*). In addition, the key enzymes in the synthesis of glycine betaine, choline dehydrogenase, and betaine-aldehyde dehydrogenase (encoded by *betAB*) were also found in strain MTB7, suggesting that both intake and biosynthesis of glycine betaine could be important to its cold and HHP adaptation.

Low-temperature and HHP environments also generate reactive oxygen species (ROSs), which are harmful to cells and of great significance to energy consumption. In the genome of strain MTB7, nine peroxidase and five superoxide dismutase genes were identified, in addition to the genes of alkyl hydroperoxide reductase and organic hydroperoxide resistance protein. These genes could help strain MTB7 scavenge intracellular ROS and thus make cells grow better under cold and/or HHP stressors.

## 4. Conclusions

Strain MTB7, isolated from the water-sediment interface of the Mariana Trench at 5900 m in depth, belongs to the genus *Shewanella*. Both higher genomic redundancy and more genes from horizontal gene transfer contributed to its large genomic size. Moreover, higher coding density is an important genomic characteristics of strain MTB7, which could increase metabolic flexibility to adapt to the hadal environments. Metabolically, strain MTB7 could utilize very few carbohydrates but various amino acids as carbon sources. The HGT-derived genes in D-amino acid and purine utilization could be important for its adaptation to the oligotrophic environment in a hadal ecosystem. Multiple electron transfer chains, including various anaerobic respiration enzymes, could allow MTB7 to thrive in deep marine water-sediment interfaces through fermentation or respiration using alternative electron acceptors. Many genes of cold shock proteins, glycine betaine transporters, and biosynthetic enzymes, as well as reactive oxygen species- scavenging proteins, could contribute to its low-temperature adaptation. In the future, genomic information obtained in the present study could be used to gain a deeper understanding of the stress-resistance strategies developed by strain MTB7. In addition, this study provides a reference for further phylogenomic, comparative genomic, and functional studies of other relevant species in the deep ocean.

## Figures and Tables

**Figure 1 microorganisms-11-00798-f001:**
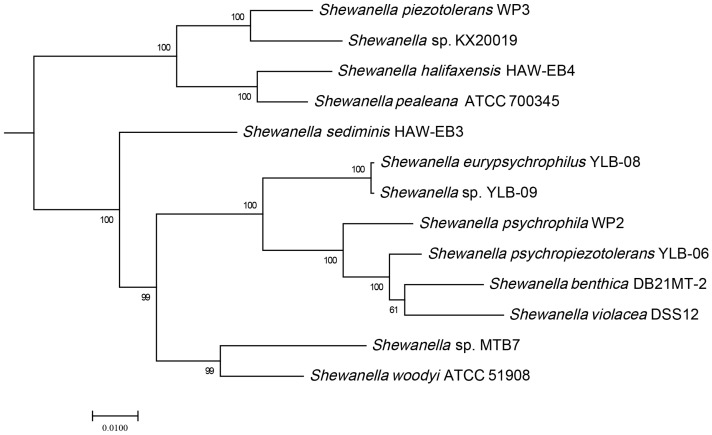
The phylogenetic tree of genus *Shewanella* based on 120 concentration proteins. This tree was constructed with all 117 genomes of *Shewanella* in the NCBI RefSeq database, and only the strain MTB7-related branch is shown. The bootstrap values were also shown in number.

**Figure 2 microorganisms-11-00798-f002:**
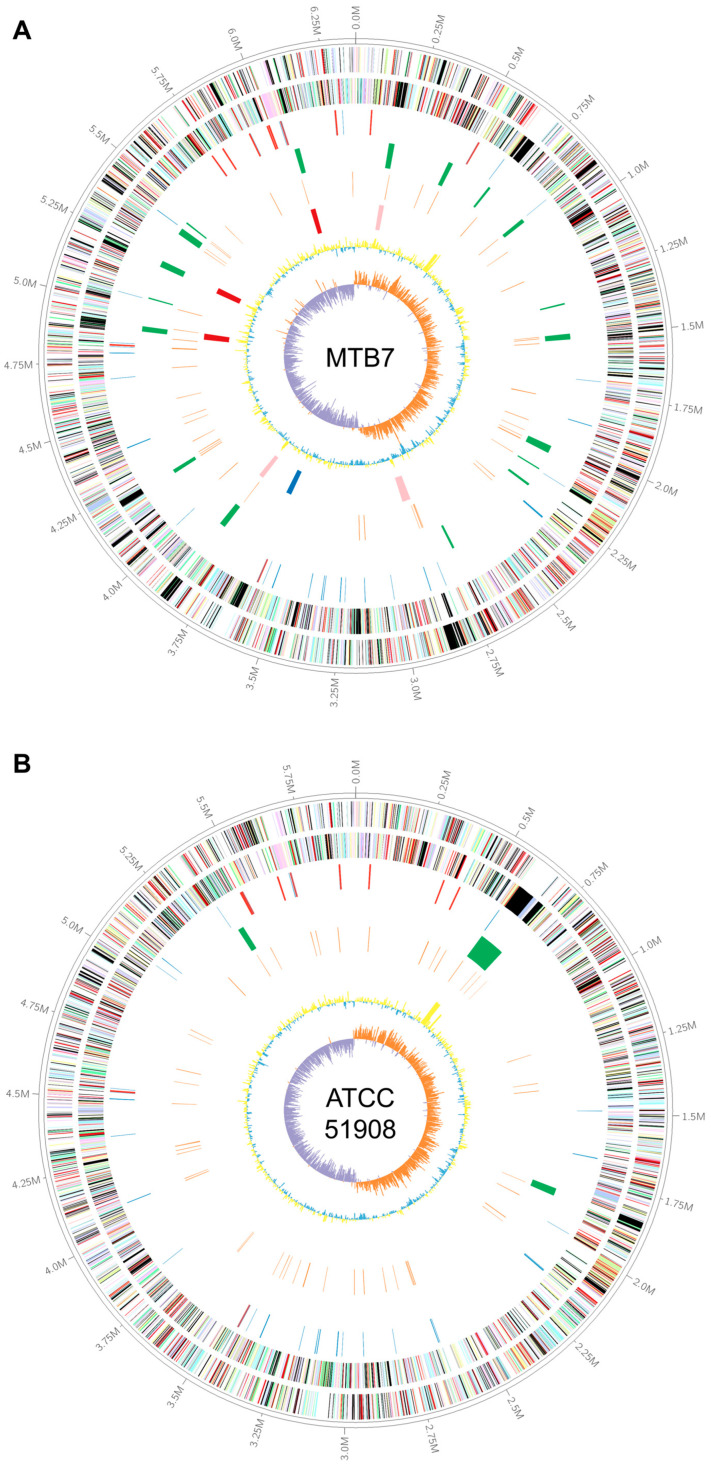
Graphical representation of the *Shewanella* sp. MTB7 (**A**) and *Shewanella woodyi* ATCC 51908 (**B**) genomes (window, 5000 bp; step, 2500 bp). Genes on the forward (shown in the outer circle) and reverse (shown in the inner circle) strands are colored according to their cluster of orthologous gene (COG) categories (except those colored in black for no hits); RNA genes are highlighted with different colors (tRNAs blue and rRNAs red); gene islands are shown in green; genes of transposases, recombinases, and integrases are shown in orange; proviruses are show in different colors (red for high qualified, pink for medium qualified, and blue for low qualified); GC content is shown in yellow/blue; and GC skew is shown in orange/purple.

**Figure 3 microorganisms-11-00798-f003:**
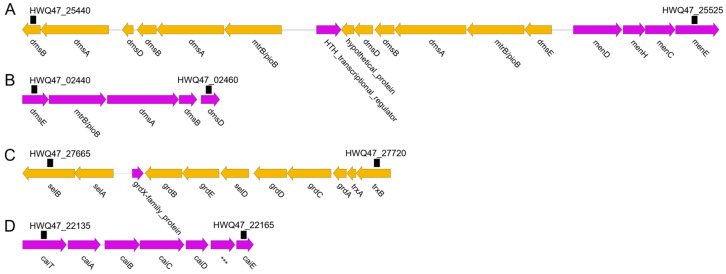
Gene clusters of type TMAO reductase (**A**,**B**), glycine reductase (**C**), and crotonobetaine reductase (**D**) in *Shewanella* sp. MTB7 genome.

**Table 1 microorganisms-11-00798-t001:** Genome features of strain MTB7.

Items	Description
Size (bp)	6,367,435
G+C content (%)	42.70
Coding sequence (%)	89.93
Total genes	5421
Protein-coding genes	5301
Genes assigned to COG	3953
rRNA operons	11
tRNA genes	127
Gene islands	18
Orthologous gene families	5198
Paralogous genes	338

## Data Availability

The GenBank/EMBL/DDBJ accession number for the genome sequence of strain MTB7 is CP085636.

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
