# Peer review of "Genomic Analysis of the Deep-Sea Bacterium Shewanella sp. MTB7 Reveals Backgrounds Related to Its Deep-Sea Environment Adaptation"

_microorganisms, 2023, doi:10.3390/microorganisms11030798_

Round 1

Reviewer 1 Report

Overall this is a solid description of the complete genome of an extremophile bacterium. Data have been submitted to public databases, and I only have minor questions/suggestions to improve the manuscript:

Line 18 “contains currently the 3rd largest genome?” Please specify that you are comparing to the genus Shewanella

19 “Metabolically, MTB7 could utilize various carbon and nitrogen sources.” à these are predictions, not experimental proof of metabolic capacities, so perhaps phrase accordingly

24 What is meant by ‘strain-specific’ here? (you define this later in methods, but the abstract needs to be understandable without referring to the methods)

40: constitutes à comprises

53: believed à are believed

85: I am wondering about the assembly method. Was it Unicycler as mentioned in the manuscript or Canu as mentioned in the Genbank submission?

Section 2.4: How many positions are in the final alignment?

166: “These pieces of evidence suggested that horizontal gene transfer (HGT) has contributed to the increase of orthologs and genome expansion of strain MTB7”. Please be more precise about what evidence and how this supports horizontal gene transfer.

178: This hypothesis sounds plausible, but how would you reconcile this with the observation of increased redundancy from line 170? Could you comment on this?

Figure 2: Could the authors provide a more detailed Legend? How do you interpret the GC skew? Any hypotheses on the shift that concerns roughly half of the genome?

Section 3.3: Regarding your comparisons of metabolic networks, did you use the same pipeline to reconstruct the network of S. woodyi? Please clarify – if different pipelines were used, this might significantly impact the comparisons.

212: This is an interesting finding, yet it is presented only vaguely. What are the identity percentages, where are these sequences found, and which strains/genera come from the deep sea? Perhaps a tree would be useful as well? To a lesser extent (since you provide more information already), this also applies to line 249.

283: Do you have any citation for this temperature range, or is it your data? Please add a source, or describe the corresponding methods and results in more detail in your manuscript.

304: remove ‘there are’

Supplementary Table S1: Perhaps add a description of COG categories, why is the first column all 0?

Supplementary Table S2: Please define ‘Closely related to Shewanella sp. MTB7’

Reviewer 2 Report

Major point

The work is well performed an of certain interest, but, from what I have seen lately for publication in Microorganisms is necessary something additional to the genome sequencing and analysis of just one bacterial strain.

Minor points

Line 141  change "starin" to strain

Line 198 change "the genes in..." to "the genes for..."

Line 199 "It suggested" to "This suggests"   "lost a mass"  to "lost many"

Line 200 "carbohydrates were deprived" to "... were scarce"

Line 204 "survival" to "survive"

Line 205 eliminate "refractory"

Line 207 should be: "phospholipids could also be a possible carbon source for MTB7"

Line 212 "acquirement" to "acquisition"

Line 215 should be "were predicted to have genes for denitrification ..."

Line 222 should be "respiration in central in the metabolism of..."

Line 232 "genes in anaerobic.." should be "genes for anaerobic"

Line 253 change "preocuppies" to "harbours the..."

Line 285 "is as an RNA.." change to "is similar to that of a RNA..."

Line 286 eliminate " combining with single-stranded ribonucleic acid/deoxyribonucleic 286 acid to" change to "promoting"

Line 290 change "top" to "high"

Line 296 change "preocuppies" to "harbours"

Line 304 eliminate "were identified"

Line 305 change "in addition with" to "in addition to"

Line 313 change "bring it" to "increase" , and "to adapt" to "to adapt to"

Line 317 change "especially" to "including"

Line 322 "stress resistant" to "stress resistance"

In Supplementary Table S1 the column B should be eliminated, only contains 0 in all cells. It should also indicate the name of the category associated to each letter.

In line 160 the author state that "strain MTB7 owns more orthologous families (5,536) than neighboring strains (from 3,532 to 5,332) (Supplementary Table S2)"

First "owns" should be changed to "has".  The number 3532 is repeated, other number should be in its place. Then, the differences 5536 to 3532 are of little significance. Of more significance would be the comparisson with the average of the 12 closely related strains in Table S1.

Line 161. The genomic islands positions should be indicated in Table S3

Table S4. A column with an indication of the family/type of provirus shoud be included.

Table S6, an additional column indicating the short notation of the genes (e.g. frdA) should be included

Reviewer 3 Report

1- The authors, in the materials and methods section, should better explain how they isolated the bacterial strain MTB7 from the samples of water-sediment then let it grow in marine broth 2216.

2- In the paragraph of materials and methods “DNA extraction, genome sequencing and assembly” The authors should first describe how the library was prepared and then how the raw data analysis was conducted, then the sentence “Raw reads were quality filtered and adapter trimmed with Trimmomatic” it must be moved.

3- Figures 1, 2, 3 must be enlarged because they cannot be seen well.

4- A graphical representation of the S. woodyi genome should also be added to figure 2 in order to better see the differences or to make a descriptive table, to be added to the text, in order to have an immediate picture of the differences between the two genomes.

Round 2

Reviewer 2 Report

The work is well performed an of certain interest, but, from what I have seen lately for publication in Microorganisms is necessary something additional to the genome sequencing and analysis of just one bacterial strain.